# Urinary *C-*Peptide to Creatinine Ratio (UCPCR) as Indicator for Metabolic Risk in Apparently Healthy Adults—A BioPersMed Cohort Study

**DOI:** 10.3390/nu15092073

**Published:** 2023-04-25

**Authors:** Sharmaine Reintar, Magdalena Pöchhacker, Anna Obermayer, Katharina Eberhard, Andreas Zirlik, Nicolas Verheyen, Dirk von Lewinski, Daniel Scherr, Barbara Hutz, Christoph W. Haudum, Thomas R. Pieber, Harald Sourij, Barbara Obermayer-Pietsch

**Affiliations:** 1Department of Internal Medicine, Division of Endocrinology and Diabetology and Endocrinology Lab Platform, Medical University of Graz, 8036 Graz, Austria; sharmaine.reintar@medunigraz.at (S.R.); a.obermayer@medunigraz.at (A.O.); barbara.hutz@medunigraz.at (B.H.); christoph.haudum@medunigraz.at (C.W.H.); thomas.pieber@medunigraz.at (T.R.P.); ha.sourij@medunigraz.at (H.S.); 2Department of Food Chemistry and Toxicology, Faculty of Chemistry, University of Vienna, 1090 Vienna, Austria; magdalena.poechhacker@univie.ac.at; 3Center for Medical Research, Core Facility Computational Bioanalytics, Medical University of Graz, 8010 Graz, Austria; katharina.eberhard@medunigraz.at; 4Department of Internal Medicine, Division of Cardiology, Medical University of Graz, 8036 Graz, Austria; andreas.zirlik@medunigraz.at (A.Z.); nicolas.verheyen@medunigraz.at (N.V.); dirk.von-lewinski@medunigraz.at (D.v.L.); daniel.scherr@medunigraz.at (D.S.)

**Keywords:** UCPCR, urinary *C-*peptide creatinine ratio, BioPersMed cohort, metabolic risk, prediabetes

## Abstract

***Background:** C*-peptide is produced in equimolar amounts with insulin from pancreatic beta cells, and thus is a fundamental biomarker for beta cell function. A non-invasive urinary *C*-peptide-to-creatinine ratio (UCPCR) has attracted attention as a biomarker for metabolic conditions. However, the UCPCR as an indicative risk predictor for prediabetes is still being investigated. Methods: We aimed to characterize UCPCRs in healthy people using American Diabetes Association (ADA) criteria and to evaluate their metabolic outcomes over time. A total of 1022 participants of the Biomarkers in Personalized Medicine cohort (BioPersMed) were screened for this study. Totals of 317 healthy with normal glucose metabolism, 87 prediabetic, and 43 diabetic subjects were included. ***Results*:** Prediabetic participants had a significantly higher UCPCR median value than healthy participants (*p* < 0.05). Dysglycaemia of healthy baseline participants was measured twice over 4.5 ± 0.9 years; 25% and 30% were detected with prediabetes during follow-ups, predicted by UCPCR both for the first (*p* < 0.05) and the second visit (*p* < 0.05), respectively. This is in good agreement with the negative predictive UCPCR value of 60.2% based on logistic regression. UCPCR levels were equal in both sexes. ***Conclusion*:** UCPCR measurements provide an indicative approach for metabolic risk, representing a potential use for prevention and monitoring of impaired glucose metabolism.

## 1. Introduction

The use of *C-*peptide as a surrogate marker for insulin to assess beta cell function in clinical practice is mainly supported by its equimolar secretion and its higher stability and half-life compared to endogenous insulin, allowing for measurements in blood and urine. Notably, interferences with administered insulin or insulin autoantibodies can be neglected. Therefore, the assessment of *C-*peptide provides a mirror of the secretory pattern of insulin in healthy individuals as well as in people with diabetes mellitus (DM) of various subtypes regardless of insulin therapy [1]. However, the practical use of the UCPCR in patients at metabolic risk, e.g., subjects with prediabetes (preDM), remains an open question.

In pancreatic beta cells, *C-*peptide (31 amino acids, molecular weight 3021 Da) is cleaved from the proinsulin molecule and stored in secretory granules. *C-*peptide emerges from the liver and enters systemic circulation. After a systemic half-life of around 35 min (compared to insulin with 5–10 min), *C-*peptide is excreted with the urine [2]. *C-*peptide is a pluripotent molecule with a number of currently poorly understood functions and hormonal activities, which makes its determination even more valuable [3]. However, some practical difficulties of blood *C-*peptide detection exist, including the need for immediate centrifugation and analysis due to its instability and short half-life. Additionally, testing is even more difficult in remote areas lacking sophisticated instruments, which requires patients to transport the blood *C-*peptide kept on ice, making it impractical and inconvenient. There are existing investigations about the correlations of 24-h urinary *C-*peptide (24-h UCP) with fasting blood *C-*peptide [4]. However, the collection procedure of 24-h urine samples is cumbersome. For analytical purposes, urinary *C-*peptide normalized to the individual concentrations of urinary creatinine as a reference molecule to correct for the variations of urine concentration allows the use of a spot urine sample.

Several studies have reported the use of the UCPCR in different applications, including differentiation, clinical monitoring, and therapy adaptation for type 1 diabetes mellitus (T1DM) [5,6,7], hyperinsulinism, and/or type 2 diabetes mellitus (T2DM) [8,9,10,11,12], as well as independently of T2DM [5,13]. Further specific applications might include the detection of clinically relevant conditions like polycystic ovary syndrome (PCOS) or gestational diabetes (GDM) in pregnant women at risk [14,15], and a potential application in postoperative monitoring of pancreas graft function [16], where a close but low threshold, and at best non-invasive, monitoring is mandatory. However, the use of the UCPCR to predict metabolic risk factors such as preDM is still being investigated. Therefore, we aim to define guiding values for the UCPCR in a precisely defined and thoroughly phenotyped population of healthy participants, allowing for an individual risk prediction for the future risk for impaired glucose metabolism from non-invasive urinary samples and the documentation of metabolic changes over time.

## 2. Materials and Methods

### 2.1. Study Design and Population

The BioPersMed cohort was designed as a single-centre prospective observational cohort study consisting of a total number of 1022 volunteers, recruited between 2010 and 2016. Subjects aged 45 years or older were included when having at least one traditional cardiovascular risk factor or manifesting T2DM, with a proportion of 26% at low cardiovascular risk according to the Framingham “Systematic COronary Risk Evaluation” (SCORE). The BioPersMed cohort study was approved by the Ethics Committee of the Medical University of Graz, Austria (EC Nr. 24-224 ex 11/12), and it was conducted in compliance with Good Clinical Practice Guidelines Procedures (GCP) and complies with the Declaration of Helsinki and the Austrian laws. Further detailed description of the study design was published previously [17].

Inclusion criteria were based on ADA definitions including a classification of preDM as a fasting plasma glucose (FPG) of 100–125 mg/dL (5.6–6.9 mmol/L), 2-h postprandial glucose (2-h PG) of 140–199 mg/dL (7.8–11.0 mmol/L) during a 75-g oral glucose tolerance test (oGTT), and hemoglobin A1c (HbA1c) of 5.7–6.4% (39–47 mmol/mol), whereas T2DM was defined by FPG ≥ 126 mg/mL (≥7.0 mmol/L), 2-h PG ≥ 200 mg/dL (≥11.1 mmol/L) during the oGTT, and HbA1c ≥ 6.5% (≥48 mmol/mol). Healthy subjects were chosen based on ADA criteria that showed neither classification of preDM nor T2DM. Among 447 total participants included in the current work, 317 were healthy participants (female n = 198 (62.5%)), (male n = 119 (37.5%)), 87 prediabetic (female n = 32 (36.8%)), (male n = 55 (63.2%)), and 43 diabetic ((female n = 15 (34.9%)), (male n = 28 (65.1%)). Subjects with T1DM were not included in the study. Importantly, participants with missing information/data were excluded from the study, as well as all patients with hormonal imbalances other than post menopause in women, renal failure, congestive heart disease, chronic respiratory problems, liver disease, malabsorption syndromes, or other severe diseases.

### 2.2. Study Procedures, Samples, and Parameters

A physical examination and anthropometric, functional metabolic, and laboratory testing of blood and urine samples were performed during screening and follow-up visits. Such analyses included liver and kidney function and electrolytes, blood counts, hormonal and metabolic data including lipid profiles, fasting blood glucose, insulin and *C-*peptide, and urinary analyses. Additionally, standardized oral glucose tolerance, insulin, and *C-*peptide tests were also performed.

After an overnight fast of at least 8 h, blood samples were drawn in the morning using a Vacuette Luer Adapter (Greiner Bio-One, Kremsmünster, Austria) in a sitting position, and urinary samples using urinary cups by Greiner Bio-One, Austria were collected as a second morning void. Following a concise protocol, samples were either immediately analysed or biobanked according to a fixed standard operating procedure and frozen at −80° Celsius in the Biobank of the Medical University Graz (biobank.medunigraz.at/en) accessed on 24 April 2023.

To determine *C-*peptide concentrations, blood and urine samples were centrifuged at 2500 relative centrifugal force (RCF) for 10 min at room temperature (RT) (20–25 °C), and 1000 RCF for 15 min at RT, respectively. Subsequently, analysis was performed using ADVIA Centaur^®^ system (Siemens Health Care, Vienna, Austria). In addition, insulin and other parameters were determined according to the conditions of good laboratory practice (GLP) and good scientific practice (GSP). Measurements of FPG, HbA1c, and other biochemical variables such as alanine aminotransferase (ALT), aspartate aminotransferase (AST), gamma-glutamyl-transferase (GGT), blood and urinary creatinine, triglycerides, cholesterol, low-density lipoprotein (LDL), and high-density lipoprotein (HDL), among others, were performed using a Cobas^®^ Analyzer (Roche Diagnostics, Penzberg, Germany). Functional data from oGTT were available for all subjects and have been included in the analysis, as well as the Homeostatic Model Assessment for Insulin Resistance (HOMA-IR), the Stumvoll and Cederholm Insulin Sensitivity Indices (ISI), and the Matsuda index (based on Otten et al. 2014) [18,19].

Additional clinical parameters were available to thoroughly characterise healthy, preDM, and T2DM groups. We included the participants’ medical history for diseases, medications, and anthropometric measurements such as weight, height, and waist and hip circumference. Body mass index (BMI) was calculated using this formula: weight (kg)/height (m^2^) (kg/m^2^). Body composition measurements were performed using dual-energy-X-ray densitometry (Lunar iDXA^®^, General Electrics, Boston, MA, USA) for parameters of total lean and total fat body mass.

### 2.3. C-Peptide Assay Characteristics

For *C-*peptide measurements using the ADVIA Centaur^®^ system, calibration and quality control for precision and accuracy of the instrument during the measurements were performed according to the manufacturer’s instructions. The analytical measurement range of the assay was 0.05–30 ng/mL and 0.50–300 ng/mL for blood and urinary *C-*peptide, respectively. Interference testing was determined using the CLSI Document EP7-P according to the Clinical and Laboratory Standards Institute (formerly NCCLS) with no known cross-reactivity to substances such as proinsulin, insulin, glucagon, calcitonin, somatostatin, or secretin. The sensitivity of the assay was described up to 30 ng/mL with a minimum detectable concentration of 0.05 ng/mL. Furthermore, the assay was reported with a precision of 3.7, 4.0, 4.1 (% CV (within run)); 3.3, 1.1, 1.0 (% CV (run-to-run)); 6.1, 5.1, 6.2 (% CV (total)) for 1.4 ng/mL, 4.9 ng/mL, 10.6 ng/mL, respectively, for serum samples, and 4.7, 4.1 (% CV (within run)); 5.1, 3.6 (% CV (run-to-run)); 8.5, 9.5 (% CV (total)) for 10.4 ng/mL and 37.1 ng/mL, respectively, for urine samples.

The accuracy standard of the assay for serum and urinary *C-*peptide measurements was within the ranges of 0.23 to 22.72 ng/mL and 0.42 to 300 ng/mL, respectively. Dilution recovery for serum and urinary *C-*peptide ranged from 84.1 to 106.3% (mean = 93.5%) and 96.6% to 119.9% (mean = 105.7%), respectively. Spike recovery for serum and urinary *C-*peptide ranged from 90.1 to 108.8% (average = 97.8%) and 96.9 to 105.0% (average = 101.7%), respectively.

### 2.4. Statistical Analysis

Assumption of normal distribution was shown with Kolmogorov–Smirnov or Shapiro–Wilk tests (*p* > 0.05 normally distributed data assumed) and Q-Q plots. The non-parametric Kruskal–Wallis tests with Bonferroni correction for multiple testing were used for testing group differences (healthy group compared to diseased groups) according to clinical baseline characteristics. Correlation studies were analysed using Spearman correlation coefficients, due to the not normally distributed data of the parameter UCPCR. Associations between categorical variables were analysed with Chi-square tests and Fisher’s exact tests. Data were presented as total number or relative frequencies, and in case of a skewed distribution, as median and interquartile range (25-percentile and 75-percentile). A *t*-test was used for normally distributed data and the Mann–Whitney U test was used for skewed data to assess differences between the two groups. A binary logistic regression model was used to evaluate predictors for preDM. A *p*-value of < 0.05 was considered as statistically significant. All tests were 2-sided with 95% confidence intervals (95% CIs). The propensity score matching was done in R 4.1.3 using package MatchIt. Each individual in the preDM group was matched to an individual in the healthy control group based on the confounding factors gender, BMI, waist, hip, and age in years on the nearest matching function. Unmatched participants were discarded. Statistical tests were performed using Statistical Package for Social Sciences (IBM SPSS) version 27.0 (SPSS Inc., Chicago, IL, USA). GraphPad Prism 8 (GraphPad Prism version 8.0.2 for Windows, GraphPad Software, La Jolla, CA, USA) was used for visualisations. The Sankey plot was designed using the online software at SankeyMATIC: Make Beautiful Flow Diagrams (sankeymatik.com), last accessed on 20 March 2023.

## 3. Results

Characteristics of all participants and defined groups (heathy, preDM, and T2DM) at baseline are shown in Table 1. We found differences between the predefined groups in almost all parameters listed, except for height and urinary creatinine values.

Out of 1022 BioPersMed participants, a total of 447 (male, n = 202 (45.2%)), (female, n = 245 (54.8%)) participants were selected for this study, including 317 healthy people with normal glucose metabolism and 87 prediabetic and 43 diabetic people. A non-normal distribution was observed between genders (Pearson Chi-square, *p*-value < 0.001). Of note, men were more likely to develop preDM (*p* < 0.001) or T2DM than women (*p* < 0.001) based on Fisher’s exact test.

Anthropometric and laboratory data of the study group presented in Table 1 showed that participants with preDM had significantly higher UCPCR, BMI, hip and waist circumference, WHR, weight, total lean mass, total fat mass, FPG, AUC glucose, fasting *C-*peptide, hbA1c, 1-h and 2-h stimulated *C-*peptide, AUC *C-*peptide, fasting blood insulin, AUC insulin, HOMA-IR, blood creatinine, and UCP compared to the healthy group (*p* < 0.05). In addition, we found a lower Matsuda index, ISI Stumvoll and ISI Cederholm in the preDM group compared to healthy people (*p* < 0.05).

Due to the higher sample size in the healthy group compared to the preDM group, we performed a propensity score matching on the subgroups. Based on our analysis using the matched data, we found no substantial differences in the descriptive statistics of the population at the baseline (see Appendix A).

Urinary *C-*peptide (*p*-value < 0.001) and urinary creatinine (*p*-value < 0.001) showed significant differences between male and female participants. However, after normalization of urinary *C-*peptide to the individual urinary creatinine values, no significant difference was observed based on Mann–Whitney U tests (*p*-value = 0.162 Figure 1). Even after propensity score matching was applied to balance the sample size between gender, differences in UCPCR values in both sexes remained statistically non-significant (*p*-value = 0.921).

Potential covariates were tested using correlation studies for both healthy and preDM groups at baseline, listed in Table 2 and Table 3, respectively. Factors influencing UCPCR have been identified such as age, BMI, hip and waist circumference, hbA1c, AUC *C-*peptide, fasting blood *C-*peptide and insulin, HOMA-IR, and blood creatinine in the healthy group (see Table 2). Notably, factors such as BMI, waist circumference, WHR, total fat mass, fasting blood glucose, *C-*peptide and insulin, AUC glucose and *C-*peptide, 1-h and 2-h stimulated blood *C-*peptide, HOMA-IR, ISI Stumvoll, ISI Cederholm, and the Matsuda index had a significant impact on the UCPCR levels in preDM subjects (see Table 3).

As shown in Table 2, we found a significant positive correlation between UCPCR and age (r = 0.119), BMI (r = 0.111), hip circumference (r = 0.114), waist circumference (r = 0.116), hbA1c (r = 0.146), fasting blood *C-*peptide (r = 0.227), AUC *C-*peptide (r = 0.126), fasting blood insulin (r = 0.111), HOMA-IR (r = 0.112), and a negative correlation with blood creatinine (r = −0.115) in the healthy group (*p* < 0.05). However, we found that UCPCR is not correlated with 1-h and 2-h stimulated blood *C-*peptide in healthy participants.

In Table 3, we found a significant positive correlation between UCPCR and BMI (r = 0.225), waist circumference (r = 0.266), WHR (r = 0.274), total fat mass (r = 0.213), fasting blood glucose (r = 0.238), AUC glucose (r = 0.241), fasting blood *C-*peptide (r = 0.324), 1-h stimulated blood *C-*peptide (r = 0.286), 2-h stimulated blood *C-*peptide (r = 0.235), AUC *C-*peptide (r = 0.311), and fasting blood insulin (r = 0.250), HOMA-IR (r = 0.261), and a negative correlation with ISI Stumvoll (r = −0.250), ISI Cederholm (r = −0.232), and the Matsuda index (r = −0.236) in the preDM group (*p* < 0.05).

Additionally, fasting blood *C-*peptide is correlated with urinary *C-*peptide in healthy people: r = 0.322 (*p* < 0.001) and in the preDM group: r = 0.304 (*p* = 0.004).

A binomial logistic regression was performed to identify the effects offasting blood glucose, C-peptide and UCPCR on the likelihood that participants have preDM. The model showed a statistically significant result (p < 0.001) in all the models shown in Table 4–The odds of having preDM is 3.419 times greater for each increase in UCPCR values and a odds ratio of 1.401 and 3.320 greater for fasting blood glucose and C—peptide. Based on this model, the best predictors for preDM are fasting blood glucose with a specificity of 87.3% and sensitivity of 83.9% and fasting blood *C-*peptide with a specificity of 73.6% and sensitivity of 59.8%, followed by UCPCR, with a specificity of 71.3% and sensitivity of 52.9%, based on univariate binary logistic regression.

In Table 5, a classification table is presented, wherein the negative predictive values of the UCPCR were calculated to represent the percentage of correctly predicted cases without the observed characteristic, compared to the total number of cases predicted as not having the characteristic. Based on our data, this is 100 × (62 ÷ (62 + 41)), which is 60.2%. This shows that, of all cases predicted as not having preDM, 60.2% were correctly predicted by the UCPCR model.

To demonstrate a practical model of the UCPCR application for individual patients, we designed an indicative gradient based on UCPCR values, and categories of healthy, preDM, and T2DM (or hyperinsulinaemic) persons (Figure 2).

Using a Sankey plot, healthy participants defined by ADA criteria with available UCPCR values were followed over time for 2.2 ± 0.5 and 4.5 ± 0.9 years for their metabolic outcome (Figure 3). Baseline healthy participants assigned to their later outcome showed a significant prediction by UCPCR values both after 2.2 ± 0.5 years at the first follow-up (*p* = 0.01) and after about 4.5 ± 0.9 years at the second follow-up (*p* = 0.01).

## 4. Discussion

In this study, we describe UCPCR values as indicative markers of an individual’s metabolic situation both at baseline and over time. We found that men were more likely to develop preDM and T2DM than women. Furthermore, we observed that UCPCR values in adults with preDM are higher than those in healthy adults, which is likely due to their impaired glucose metabolism. Therefore, the UCPCR might be a useful risk gradient for a personalized, non-invasive assessment of metabolic risk.

By normalizing urinary *C-*peptide to the urinary creatinine concentration, the UCPCR accounts for variations in individual urinary concentrations. Although it is known that the creatinine excretion rate is constant throughout the day, gender differences based on muscle mass greatly influenced the creatinine values. In this case, as UCPCR values were obtained using creatinine concentration, it was expected that UCPCR values would be higher in females than males because of the lower urinary creatinine excretion rates. Therefore, we evaluated the impact of gender difference on subgroups defined by ADA criteria. In our study, UCPCR values did not differ between women and men, in contrast to the findings of Thomas et al., 2012, in which they found a 1.48-fold higher UCPCR values in women than in men [20], and to urinary *C-*peptide and creatinine alone. Further, specific individual DXA-derived body composition data were used to underpin this fact and to enable the use of uniform thresholds and transition zones of UCPCR values for women and men.

In our study, we found a significant positive correlation between UCPCR and age, BMI, hip and waist circumference, fasting blood *C-*peptide, AUC *C-*peptide, fasting blood insulin, HOMA-IR, and a negative correlation with blood creatinine in the healthy group. This result agrees with Oram et. al., (2013) [13] and Katte J. et. al., (2020) [9]. Hence, UCPCR correlated to fasting serum insulin and *C-*peptide in healthy groups suggests the utilization of the UCPCR in population-based epidemiological studies to assess insulin secretion and its metabolic outcome (e.g., dysglycaemia or preDM) without complex blood extraction and healthcare assistance for collecting samples.

To answer the question of whether the UCPCR can be an indicative marker for metabolic risk (e.g., preDM) for individuals healthy at baseline, we performed a binary logistic regression analysis, and further calculated the sensitivity, specificity, and negative predictive value of the UCPCR as a predictor. Interestingly, we found that of all cases predicted to not develop preDM, 60.2% were correctly predicted by UCPCR. Therefore, the unpredicted percentage of the population (39.8%) might be at risk of developing preDM. To prove this, we investigated their metabolic outcomes during two biannual follow-ups. Importantly, this is one of the strengths of our cohort study, which is designed as longitudinal. Based on our study, 25% and 30% of the healthy-at-baseline participants progressed to preDM during the first follow-up and second follow-up, respectively. Additionally, 1% of the healthy-at-baseline participants progressed to T2DM during the second follow-up. This finding indicates that UCPCR values are significant predictors of progression to a dysglycaemic state. Notably, the investigated group was representative of healthy elderly people without any other disease with a mean weight within the normal range. Interestingly, individual patients showed a considerable change between the metabolic categories, demonstrating individual changes in insulin resistance and subsequent glycaemic category.

Despite some suggestions for a use of UCPCR measurements after a meal [9], we have shown a relatively small variation in fasting UCPCR in our study, which might add to the standardization of the parameter [21]. Because of the variation of the *C-*peptide secretion in response to the evening meal which accumulates in an overnight urine sample, second-void fasting urine was collected due to its lower variance than first-void urine. The second void urine has been found to correlate best with 24-h UCP values in a study by McDonald et al. [22], and might therefore be the urine of choice for future measurements. In addition, UCPCR determination is a very practical, non-invasive test, which provides stability for three days at room temperature in boric acid as a preservative, in contrast to blood *C-*peptide collected in an EDTA tube, which is only stable for 24 h at RT [22].

Importantly, kidney deterioration in DM might not be a problem for the calculation and clinical use of UCPCR values, as demonstrated in a recently published study that refers to 85 T2DM patients with reduced renal function [10]. Nevertheless, all participants in the current study had normal creatinine values. Whereas UCPCR values in T2DM patients [11] have been recently reported in a systematic review (including three large studies on T2DM) to have a sensitivity, specificity, and diagnostic odds ratio (DOR) of 92.8% (84.2–96.9%), 81.6% (61.3–92.5%), and 56.9 (31.3–103.5%), respectively, our intention was not to identify T2DM per se, but rather to test a monitoring tool for individual patients at risk, which can potentially be used at home in addition to common diagnostic procedures.

For clinical use, a number of potential indications and approaches for diagnostic applications have already been suggested. These applications cover topics from pediatric measurements [5,12] to pregnant women at risk for GDM to elderly people with hyper-insulinemic obesity [12], and have been validated as a reproducible alternative to serum *C-*peptide, e.g., in patients with T2DM [23]. Additionally, Oram et. al., 2014, reported that UCPCR sent from home can provide an alternative valuable measurement of the status of the patient’s undergone islet transplant [16]. Moreover, the convenience of using non-invasive urine sampling has already benefited animal studies in its use as a marker of nutritional status in macaques [24].

Limitations of the study were the lack of first morning urinary voids and 24-h urinary sampling to compare with other material. Furthermore, there are additional guidelines for diabetes and non-diabetes definitions of other origin in use. However, we believe that these guiding measurements are of practical value for clinicians. In addition, the BioPersMed study was designed for apparently healthy participants with one out of various cardiovascular risk factors to follow them for their metabolic and cardiovascular outcomes. Therefore, the cohort was clearly not designed for or meant to be used for a specific T2DM study. Additionally, UCPCR measurement with renal impairment was not evaluated in this study. The diabetic patients were under treatment.

On the other hand, the study has important strengths; by using additional clinical data of the BioPersMed participants such as body composition and functional tests, we were able to describe a very careful and detailed phenotyping, and the study was designed as longitudinal to follow and monitor patient outcomes.

## 5. Conclusions

UCPCR measurement showing a good negative predictive value and specificity can be a promising potential predictive biomarker for people at risk of metabolic conditions, providing an early insight regarding their beta cell function. UCPCR guiding values are normalized, allowing their detection in a urine spot specimen, and are independent to muscle mass, hence equal between genders, indicating its practical use in the laboratory for developing a single reference range. Furthermore, the UCPCR provides a simple, easy, time-efficient, and convenient test to perform, in comparison to blood *C-*peptide and 24-h urine *C-*peptide. Applications of the UCPCR are not limited, but may be proposed in the future for the use of this personalized low-threshold method, e.g., for hyperinsulinemia in obese people, PCOS subjects, and those experiencing GDM. Clinical applications and new insights are important to establish the potential of UCPCR as a low-threshold screening and monitoring parameter for metabolic conditions. This study suggests that the UCPCR can be a tool for preventive management, particularly for metabolic conditions such as elevated endogenous insulin.

## Figures and Tables

**Figure 1 nutrients-15-02073-f001:**
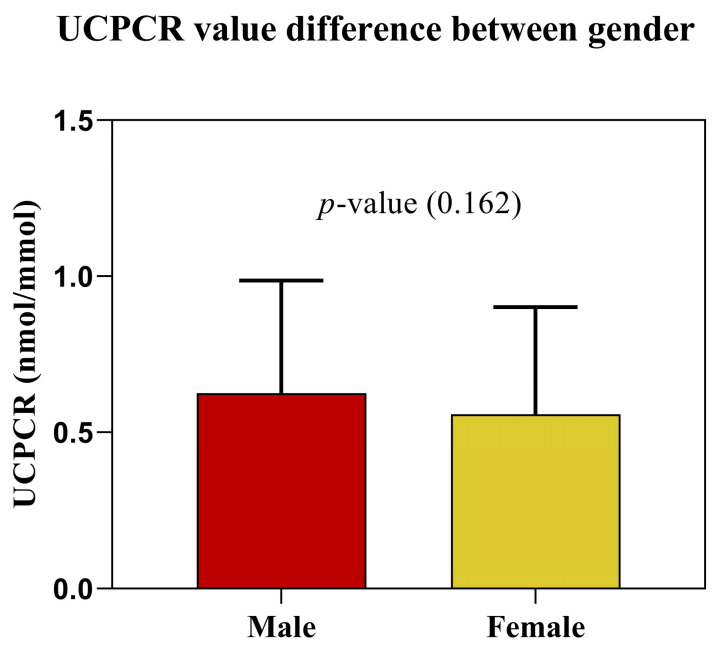
After normalization for individual urinary creatinine, UCPCR values in women and men did not show significant differences (*p*-value = 0.162).

**Figure 2 nutrients-15-02073-f002:**
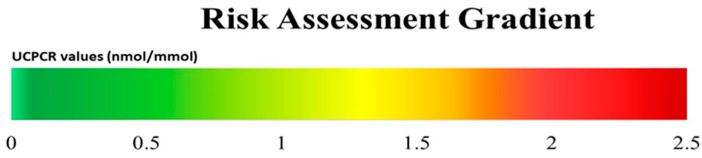
Proposed metabolic risk assessment gradient for individual UCPCR (nmol/mmol) values in relation to glycaemic categories: healthy individuals (green) in our cohort, as well as preDM (yellow) and T2DM/hyperinsulinism (red) groups according to ADA criteria, showing transition zones for each category.

**Figure 3 nutrients-15-02073-f003:**
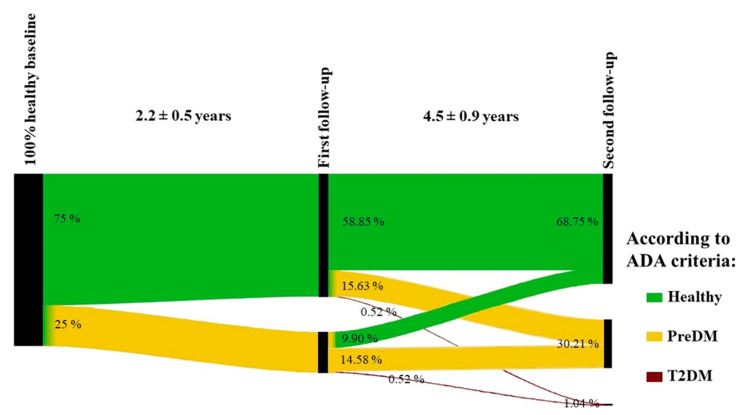
Schematic representation of the time course of the participants. Individuals that were 100% healthy at the baseline (n = 192) were followed over 2.2 ± 0.5 and 4.5 ± 0.9 years for their metabolic outcome.

**Table 1 nutrients-15-02073-t001:** Baseline characteristics of the study group. Values are represented as median ± interquartile range (IQR, 25–75th percentile). NA means not applicable due to non-statistically significant values from Kruskal–Wallis test. BMI: body mass index; WHR: waist-to-hip ratio; AUC: area under the curve; HbA1c: hemoglobin A1c; HOMA-IR: Homeostatic Model Assessment for Insulin Resistance; ISI: Insulin Sensitivity Index; UCP: urinary *C-*peptide; UCR: urinary creatinine; UCPCR: urinary *C-*peptide-to-creatinine ratio. HbA1c: hemoglobin A1c [20–42 mmol/mol]; fasting blood glucose (70–100 mg/dL); fasting blood *C-*peptide [0.78–1.89 ng/mL]; fasting blood insulin (3–25 mU/L); blood creatinine (female: 0.50–0.90 mg/dL, male: 0.70–1.20 mg/dL). Numbers in bold are considered statistically significant (*p* value < 0.05).

Variables	All N = 447	Healthy N = 317	PreDM N = 87	T2DM N = 43	*p*-Value	Healthy vs. preDM (*p*-adj.)	Healthy vs. T2DM (*p*-adj.)	PreDM vs. T2DM (*p*-adj.)
**Age (years)**	57 (51–63)	56 (51–62)	59 (53–66)	61 (58–67)	**<0.001**	**0.021**	**0.002**	0.728
**BMI (kg/m^2^)**	25.90 (23.67–29.05)	24.84 (22.75–27.50)	27.80 (25.60–30.80)	30.80 (26.10–33.90)	**<0.001**	**<0.001**	**<0.001**	0.111
**Hip** **circumference (cm)**	99 (92–106)	96 (90–103)	103 (95–109)	108 (102–118)	**<0.001**	**<0.001**	**<0.001**	**0.007**
**Waist** **circumference (cm)**	90 (80–100)	87 (77–95)	96 (90–105)	107 (97–116)	**<0.001**	**<0.001**	**<0.001**	**0.035**
**WHR**	0.92 (0.86–0.97)	0.90 (0.84–0.95)	0.96 (0.90–1.01)	0.98 (0.92–1.02)	**<0.001**	**<0.001**	**<0.001**	0.849
**Weight (kg)**	77 (66–86)	72 (63–83.50)	82 (74–88)	88 (80–103)	**<0.001**	**<0.001**	**<0.001**	0.124
**Height (cm)**	170 (164–178)	169 (164–176.50)	173 (165–179)	172 (165–178)	0.138	NA	NA	NA
**Total lean mass (kg)**	47.70 (40.75–57.50)	44.85 (39.85–56.27)	52.92 (43.77–57.93)	56.78 (45.29–62.98)	**<0.001**	**<0.001**	**<0.001**	0.504
**Total fat mass (kg)**	24.67 (19.48–31.60)	23.23 (18.32–28.88)	28.26 (22.15–36.25)	31.33 (27.07–37.66)	**<0.001**	**<0.001**	**<0.001**	0.190
**Fasting blood glucose (mg/dL)**	91 (86–100)	88 (84–93)	104 (100–108)	140 (123–164)	**<0.001**	**<0.001**	**<0.001**	**0.029**
**AUC glucose**	14,265 (12,180–17,805)	13,095 (11,610–15,008)	18,795 (16,815–20,730)	28,110 (24,660–34,470)	**<0.001**	**<0.001**	**<0.001**	**0.001**
**HbA1c (mmol/mol)**	37 (35–39)	36 (34–38)	39 (37–42)	52 (44–59)	**<0.001**	**<0.001**	**<0.001**	**<0.001**
**Fasting blood *C-*peptide (ng/mL)**	1.34 (1.02–1.98)	1.19 (0.93–1.58)	1.98 (1.44–2.58)	2.55 (1.77–3.35)	**<0.001**	**<0.001**	**<0.001**	0.142
**1-h stimulated blood *C-*peptide** **(ng/mL)**	6.38 (5.00–8.46)	6.15 (4.59–7.98)	7.82 (6.20–10.34)	6.01 (4.54–8.15)	**<0.001**	**<0.001**	1.000	**<0.001**
**2-h stimulated blood *C-*peptide (ng/mL)**	5.56 (3.90–8.01)	4.77 (3.59–6.56)	7.94 (5.74–11.07)	8.14 (6.09–9.75)	**<0.001**	**<0.001**	**<0.001**	1.000
**AUC *C-*peptide**	621 (477–788)	576 (456–721)	787 (620–962)	686 (529–875)	**<0.001**	**<0.001**	**0.020**	0.114
**Fasting blood insulin (mU/L)**	9.20 (6.10–13.50)	7.90 (5.00–11.35)	11.90 (9.20–19.00)	16.10 (10.10–22.70)	**<0.001**	**<0.001**	**<0.001**	0.979
**AUC insulin**	5624 (3503–9272)	4968 (3303–8062)	7872 (4587–14,186)	6275 (3962–9389)	**<0.001**	**<0.001**	0.215	0.637
**HOMA-IR**	2.11 (1.35–3.32)	1.72 (1.10–2.50)	3.09 (2.28–4.81)	5.07 (3.60–9.22)	**<0.001**	**<0.001**	**<0.001**	**0.047**
**HOMA-beta (%)**	110.77 (76.91–161.05)	117 (82.87–163.39)	109.13 (84.65–178.05)	75.27 (40.91–115.41)	**<0.001**	1.000	**0.001**	**0.002**
**ISI Stumvoll**	0.09 (0.07–0.11)	0.10 (0.08–0.11)	0.08 (0.01–0.09)	0.06 (0.04–0.09)	**<0.001**	**<0.001**	**<0.001**	0.809
**ISI Cederholm**	50.78 (35.91–66.21)	57.08 (46.17–73.11)	34.87 (25.27–47.31)	20.28 (15.58–26.81)	**<0.001**	**<0.001**	**<0.001**	**0.000**
**Matsuda index**	5.06 (3.16–8.27)	6.17 (4.20–9.38)	3.09 (1.72–4.88)	2.18 (1.28–3.35)	**<0.001**	**<0.001**	**<0.001**	0.056
**Blood creatinine (mg/dL)**	0.86 (0.76–0.98)	0.84 (0.75–0.96)	0.93 (0.83–1.05)	0.86 (0.76–0.96)	**<0.001**	**<0.001**	1.000	0.133
**UCP (nmol/L)**	5.43 (2.98–9.73)	4.64 (2.60–8.00)	8.24 (4.73–11.85)	10.83 (8.31–15.43)	**<0.001**	**<0.001**	**<0.001**	0.071
**UCR (mmol/L)**	11.23 (6.72–15.38)	10.96 (6.10–15.38)	11.85 (7.69–16.00)	11.93 (8.49–16.09)	0.287	NA	NA	NA
**UCPCR (nmol/mmol)**	0.59 (0.33–0.93)	0.49 (0.30–0.81)	0.76 (0.46–1.10)	0.92 (0.59–1.80)	**<0.001**	**<0.001**	**<0.001**	0.269

**Table 2 nutrients-15-02073-t002:** Correlation between UCPCR and the above variables in healthy individuals. BMI: body mass index; WHR: waist-to-hip ratio; AUC: area under the curve; HbA1c: hemoglobin A1c; HOMA-IR: Homeostatic Model Assessment for Insulin Resistance; ISI: Insulin Sensitivity Index; UCP: urinary *C-*peptide; UCR: urinary creatinine; UCPCR: urinary *C-*peptide-to-creatinine ratio. Numbers in bold are considered statistically significant (*p* value < 0.05).

Variables	r	*p*-Value
**Age (years)**	0.119	**0.034**
**BMI (kg/m^2^)**	0.111	**0.048**
**Hip circumference (cm)**	0.114	**0.043**
**Waist circumference (cm)**	0.116	**0.039**
**WHR**	0.017	0.760
**Weight (kg)**	0.053	0.345
**Height (cm)**	−0.036	0.527
**Total lean mass (kg)**	−0.008	0.888
**Total fat mass (kg)**	0.083	0.141
**Fasting blood glucose (mg/dL)**	0.095	0.090
**AUC glucose**	0.019	0.735
**HbA1c (mmol/mol)**	0.146	**0.009**
**Fasting blood *C-*peptide (ng/mL)**	0.227	**<0.001**
**1-h stimulated blood *C-*peptide (ng/mL)**	0.098	0.081
**2-h *C-*peptide (ng/mL)**	0.066	0.238
**AUC *C-*peptide**	0.126	**0.025**
**Fasting blood insulin (mU/L)**	0.111	**0.047**
**AUC insulin**	0.073	0.196
**HOMA-IR**	0.112	**0.047**
**HOMA-beta (%)**	0.104	0.066
**ISI Stumvoll**	−0.078	0.164
**ISI Cederholm**	−0.029	0.604
**Matsuda index**	−0.101	0.072
**Blood creatinine (mg/dL)**	−0.115	**0.041**

**Table 3 nutrients-15-02073-t003:** Correlation between UCPCR and the above variables in preDM group. BMI: body mass index; WHR: waist-to-hip ratio; AUC: area under the curve; HbA1c: hemoglobin A1c; HOMA-IR: Homeostatic Model Assessment for Insulin Resistance; ISI: Insulin Sensitivity Index; UCP: urinary *C-*peptide; UCR: urinary creatinine; UCPCR: urinary *C-*peptide-to-creatinine ratio. Numbers in bold are considered statistically significant (*p* value < 0.05).

Variables	r	*p*-Value
**Age (years)**	0.045	0.680
**BMI (kg/m^2^)**	0.225	**0.036**
**Hip circumference (cm)**	0.087	0.426
**Waist circumference (cm)**	0.266	**0.013**
**WHR**	0.274	**0.010**
**Weight (kg)**	0.143	0.186
**Height (cm)**	−0.060	0.579
**Total lean mass (k)]**	0.024	0.825
**Total fat mass (kg)**	0.213	**0.047**
**Fasting blood glucose (mg/dL)**	0.238	**0.027**
**AUC glucose**	0.241	**0.024**
**HbA1c (mmol/mol)**	0.011	0.923
**Fasting blood *C-*peptide (ng/mL)**	0.324	**0.002**
**1-h stimulated blood *C-*peptide (ng/mL)**	0.286	**0.012**
**2-h *C-*peptide (ng/mL)**	0.235	**0.029**
**AUC *C-*peptide**	0.311	**0.003**
**Fasting blood insulin (mU/L)**	0.250	**0.020**
**AUC insulin**	0.141	0.192
**HOMA-IR**	0.261	**0.015**
**HOMA-beta (%)**	0.187	0.083
**ISI Stumvoll**	−0.250	**0.020**
**ISI Cederholm**	−0.232	**0.030**
**Matsuda index**	−0.236	**0.028**
**Blood creatinine (mg/dL)**	−0.203	0.060

**Table 4 nutrients-15-02073-t004:** Univariate binary logistic regression model to predict likelihood of preDM based on fasting blood glucose, *C-*peptide, and UCPCR.

Variable	Beta	Standard Error	Odds Ratio	95% CI Lower	95% CI Upper	*p*-Value
**Fasting blood glucose**	0.337	0.050	1.401	1.270	1.546	<0.001
**Fasting blood *C-*peptide**	1.200	0.260	3.320	1.996	5.521	<0.001
**UCPCR**	1.229	0.366	3.419	1.668	7.009	<0.001

**Table 5 nutrients-15-02073-t005:** Classification table based on binary logistic regression model.

Classification Table			
Observed	Predicted		Percentage Correct (%)
	Healthy	PreDM	
**Healthy**	62	25	71.3
**PreDM**	41	46	52.9

## Data Availability

Data are available upon request. A proposal must be submitted to the corresponding authors of this manuscript to obtain access (barbara.obermayer@medunigraz.at).

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
