# Peer review of "Urinary C-Peptide to Creatinine Ratio (UCPCR) as Indicator for Metabolic Risk in Apparently Healthy Adults—A BioPersMed Cohort Study"

_nutrients, 2023, doi:10.3390/nu15092073_

Round 1

Reviewer 1 Report

Dear authors, a good paper and can be accepted in my view, however the presentation can be better for instance in the abstract it is better to make subheadings, with introduction, methods, results and conclusion, that is easier for a reader. Also in your discussion the  presentation can be better. Furtheron nice article. I found only two typo's, LIne 58:  an "m" too much after instruments   and in line 329 the "c" is missing in dysglycaemic state

Author Response

Reviewers´ comments and suggestions for authors:

Dear authors, a good paper and can be accepted in my view, however the presentation can be better for instance in the abstract it is better to make subheadings, with introduction, methods, results and conclusion, that is easier for a reader. Also in your discussion the  presentation can be better. Furtheron nice article. I found only two typo's, LIne 58:  an "m" too much after instruments   and in line 329 the "c" is missing in dysglycaemic state

Dear Reviewer 1,

First of all, we, the authors would like to thank you for giving your valuable time and feedback to our manuscript. (Manuscript ID: nutrients-2341463)

The manuscript has been revised with minor changes based on your important feedback on the manuscript.

Minor changes based on your generous comments are: 

  1. We added subheadings in the abstract such as background (line 22), methods (line 25), results (line 29), and conclusion (line 34).
  2. We changed p-value representation in the abstract in order to follow the same format representation (line 32-33).
  3. The typo errors was already corrected (please see line 58 and line 333).
  4. We restructured and added some relevant words in the discussion section of the manuscript for better representation. (line 296-299, line 313-319, line 331)

Additional changes are:

  1. We changed min to minutes under introduction section (line 53).
  2. We shortened the introduction in a way that it does not distract our main objective (line 66-74).
  3. We changed a word from DM to T2DM to avoid confusion with T1DM (see table 1 heading).
  4. We removed some parameters such as GGT in table 1,2,3.
  5. We removed the laboratory reference range in the table (see table 1), instead we added it to the legend below table 1 (see line 192-193) to avoid loaded data in the presented table
  6. We reformatted table 1 for better representation.
  7. We removed some text under result section which is already obvious in the table (line 196-197), and rephrase some paragraph and shortened it (see line 215-218)
  8. We changed and shortened the text of correlation studies by putting only one p value < 0.05, for table 2 and table 3 (line 235-239, and line 246-252)
  9. We added a (%) sign in table 5.
  10. We added or changed some words into its abbreviation form (line 60-61, line 343).
  11. We checked, deleted, adjusted references, which is mistakenly mentioned twice (line 363, line 365, line 463-464, and line 493-494).
  12. We reformulated the conclusion section in a concise way (line 380-387).

We hope that we have met your expectations based on the changes made and we are looking forward to your positive response and approval!

With best regards,

Sharmaine and the co-authors

Reviewer 2 Report

Thank you for the opportunity to review this article and I want to emphasize from the beginning that I appreciate the work done by your team to conduct this study and to develop the article. Regarding the manuscript sent, I would have some suggestions. The introduction is too long, has too many citations from the bibliography and distracts from the main purpose of the study.

The applied statistical tests are very numerous and the results tables are very loaded with data, some data being even useless.

I would suggest that the results be presented in tables that are easier to decipher. Also, I consider it unnecessary for the obvious results from the graphs and tables to be reported once again in the text.

The conclusions should be formulated more clearly and concisely.

Author Response

Reviewers´ comments and suggestions for authors:

Thank you for the opportunity to review this article and I want to emphasize from the beginning that I appreciate the work done by your team to conduct this study and to develop the article. Regarding the manuscript sent, I would have some suggestions. The introduction is too long, has too many citations from the bibliography and distracts from the main purpose of the study.

The applied statistical tests are very numerous and the results tables are very loaded with data, some data being even useless.

I would suggest that the results be presented in tables that are easier to decipher. Also, I consider it unnecessary for the obvious results from the graphs and tables to be reported once again in the text.

The conclusions should be formulated more clearly and concisely.

Response to the comments:

Dear Reviewer 2,

First of all, we, the authors would like to thank you for giving your valuable time and feedback to our manuscript. (Manuscript ID: nutrients-2341463)

The manuscript has been revised with minor changes based on your important feedback on the manuscript.

Minor changes based on your generous comments are: 

  1. We shortened the introduction in a way that it does not distract our main objective (line 66-74).
  2. We changed a word from DM to T2DM to avoid confusion with T1DM (see table 1 heading).
  3. We removed some parameters such as GGT in table 1,2,3.
  4. We removed the laboratory reference range in the table (see table 1), instead we added it to the legend below table 1 (see line 192-193) to avoid loaded data in the presented table
  5. We reformatted table 1 for better representation.
  6. We removed some text under result section which is already obvious in the table (line 196-197), and rephrase some paragraph and shortened it (see line 215-218)
  7. We changed and shortened the text of correlation studies by putting only one p value < 0.05, for table 2 and table 3 (line 235-239, and line 246-252)
  8. We reformulated the conclusion section in a concise way (line 380-387).

Additional changes are:

  1. We added subheadings in the abstract such as background (line 22), methods (line 25), results (line 29), and conclusion (line 34).
  2. We changed p-value representation in the abstract in order to follow the same format representation (line 32-33).
  3. We changed min to minutes under introduction section (line 53).
  4. The typo errors was already corrected (please see line 58 and line 333).
  5. We added a (%) sign in table 5.
  6. We restructured and added some relevant words in the discussion section of the manuscript for better representation. (line 296-299, line 313-319, line 331)
  7. We added or changed some words into its abbreviation form (line 60-61, line 343).
  8. We checked, deleted, adjusted references, which is mistakenly mentioned twice (line 363, line 365, line 463-464, and line 493-494).

We hope that we have met your expectations based on the changes made and we are looking forward to your positive response and approval!

With best regards,

Sharmaine and the co-authors